# Model-Eliciting Activities: Pre-Service Teachers' Perceptions of Integrated STEM

Cathrine Maiorca [1,*] , Jacob Martin [1] , Megan Burton [2,*] , Thomas Roberts [3] and L. Octavia Tripp [2]

1 School of Teaching, Learning, and Educational Science, Oklahoma State University, Stillwater, OK 74078, USA; jacob.martin14@okstate.edu
2 Department of Curriculum and Teaching, Auburn University, Auburn, AL 36849, USA; tripplo@auburn.edu
3 School of Inclusive Teacher Education, Bowling Green State University, Bowling Green, OH 43403, USA; otrober@bgsu.edu
* Correspondence: cat.maiorca@okstate.edu (C.M.); meb0042@auburn.edu (M.B.)

**Abstract:** This study examines how experiencing model-eliciting activities (MEAs) influenced elementary pre-service teachers' (PSTs) perceptions of an engineering-based approach to integrated STEM. The participants included 17 elementary PSTs from large public universities located in the southeastern and western regions of the United States. The participants engaged in MEA engineering-based integrated STEM learning experiences. The data included open-ended reflections about the experience. The reflections were coded deductively using the elements of the Equity-Oriented STEM Literacy Framework: dispositions, applicability and utility, empowerment, critical thinking and problem solving, identity development, and empathy. The findings indicate that when PSTs use engineering to teach mathematics and science through MEAs and approach integrated STEM with an equity focus, they increase their knowledge about the applicability and utility of STEM while simultaneously developing their identities as STEM teachers; this positively influences their dispositions towards STEM and empowers them to be teachers of STEM.

**Keywords:** integrated STEM; elementary pre-service teachers; engineering education; equity; model-eliciting activities

## 1. Introduction

Science, technology, engineering, and mathematics (STEM) education remains essential in a society more dependent on STEM innovations than ever. Individuals must be STEM-literate to function in an increasingly STEM-driven world [1] so that they can be informed and able to contribute to society [2]. Elementary teachers are the first formal experience that students have in STEM, and they have the opportunity to build on children's natural curiosity and interest in STEM [3]. However, elementary school teachers are the least prepared to teach STEM, especially in engineering [4]. This study examines how experiencing model-eliciting activities (MEAs) influences elementary pre-service teachers' (PSTs) perceptions of an engineering-based approach to integrated STEM.

### 1.1. Integrated STEM

Despite the importance of STEM to our society, there is no one agreed upon definition for STEM or integrated STEM [5]. The acronym STEM can refer to a single STEM discipline and is most commonly associated with science [2]. However, real-world problems cannot be solved using knowledge from a single discipline [5–8]. Stohlmann and others [8] defined integrated STEM as the organic combination of STEM into one activity or problem. Integrated STEM learning is defined by Moore et al. [5] as the empowering of students to collaborate and utilize content or content area practices from at least two content areas of STEM. Bybee [2] suggested that integrated STEM must include engineering because of its focus on problem solving and real-world application. Bybee [2] argued that integrated

STEM education should be based in a real-world context, where students learn about the engineering design process and 21st-century skills, "apply knowledge and skills to real-life situations," and make connections "among the STEM disciplines and state standards" (p. 117). In Mohr-Schroeder et al. [9], the authors described the characteristics of integrated STEM; it should be student-centered, integrate at least two of the STEM disciplines, focus on collaboration, and share cross-disciplinary STEM practices. In this study, integrated STEM refers to collaboratively solving real-world problems using multiple STEM disciplines, which include engineering, with the goal of increasing STEM literacy. STEM literacy is the "dynamic process and ability to apply, question, collaborate, appreciate, engage, persist, and understand the utility of STEM concepts and skills to provide solutions for STEM-related personal, societal, and global challenges that cannot be solved using a single discipline" [9], p. 33. One way to do this is through project-based learning (PBL) strategies, such as model-eliciting activities.

*1.2. Model-Eliciting Activities and Engineering Education*

Project-based learning (PBL) relies heavily on authentic problems that highlight real-world applications and simultaneously allow students to learn knowledge and skills [10]. Project-based engineering activities in the elementary classroom can occur through integrated STEM MEAs [11,12]. MEAs are in-class activities that create authentic learning opportunities by challenging students to provide solutions to open-ended, real-world problems [13,14]. Therefore, by solving MEAs, students are experiencing problem-solving practices similar to those used in professional environments [13]. Because MEAs are open-ended, students can solve problems in ways that build on their personal expertise and empower them [15]. Researchers have found that students who were traditionally considered "low-performers" performed well on MEAs [13,14]. One benefit of implementing MEAs is that student reasoning is observable throughout the problem-solving process [16].

MEAs include the construction of both abstract mathematical models and physical models [17], which makes them a natural bridge between mathematical modeling and integrated STEM education [12,18]. MEAs have been used in mathematics and engineering education and are directly correlated to the K-12 framework for quality engineering [19]. MEAs provide hands-on, real-world problem solving in tangible, relevant ways that can enhance the students' perceptions of the practical applications of STEM [20].

MEAs are developed using six design principles: model construction, reality principle, self-assessment, model documentation, generalizability, and prototype principles [13,16,18,21] (see Table 1).

**Table 1.** Description of MEA design principles.

| MEA Design Principles | Descriptions |
| --- | --- |
| Model Construction | Students develop and use a model to find their solutions to problems. |
| Reality | Problems are based in a real-life context and are meaningful to students. |
| Self-assessment | Students can test their solutions to the problems and should be able to determine whether their solution satisfies a problem's constraints. |
| Model Documentation | Students can communicate their problem-solving process to others. |
| Generalizability | Solutions are sharable with others and their models can be applied to different solutions. |
| Prototype | Solutions to MEAs are as simple as possible yet solve complex problems and help students make sense of related problems. |

While completing MEAs, the students engage in the engineering design process as they design, test, and revise their models [6]. aligns with the model construction principle and the self-assessment principle as the students design, create, and reflect on their model solutions to the real-world problem they are solving. The engineering design process is represented in Figure 1.

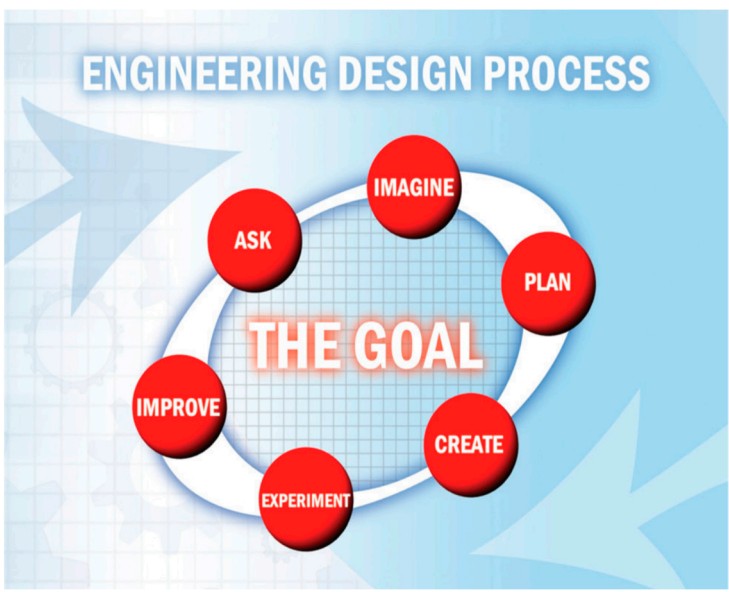

**Figure 1.** The National Aeronautics and Space Administration engineering design represented as a cyclical process with six elements [22].

MEAs are best implemented with students using cooperative learning [23]. These activities are grounded in a real-world context that is shared with students, traditionally through some form of media, that connects to their out-of-school lives [12,24]. The context-building activity is followed by open-ended readiness questions, which are usually answered individually and as a whole group. This is followed by students being given the problem statement and collaborative work time. When students finish working on the problem statement, they present their solutions to the class. Then, they are given time to revise their solutions.

### 1.3. Taking an Equitable Approach to Integrated STEM

When integrated STEM is situated around the Equity-Oriented STEM Literacy Framework, all learners, especially those historically marginalized and excluded from STEM, can see themselves as valuable members of the STEM community [25]. This framework (see Figure 2) argues that all learners must have access to and the opportunity to engage in high-quality STEM experiences. In high-quality STEM learning experiences, learners engage in hands-on, student-centered tasks that highlight the applicability and utility of STEM, fostering the development of STEM identities and positive dispositions through problem solving and critical thinking. When centering integrated STEM in the Equity-Oriented STEM Literacy Framework, MEAs would be considered high-quality STEM learning experiences. This is due to their hands-on, student-centered nature, in which students learn about how STEM can be used to solve problems for someone else. In prior research, the authors have shown that when PSTs participate in MEAs as learners they experience a positive shift in their dispositions towards STEM [26]. By engaging in high-quality STEM learning experiences, students are empowered to be societal change agents that can disrupt the systems of power and oppression that historically exclude populations from STEM.

Each component of the framework is important to analyze; when the components work simultaneously, the framework suggests that people will be empowered to be societal change agents that disrupt systems of oppression and privilege. We operationalize the six components as follows. Jackson et al. [25] state that the applicability and utility of STEM refer to the importance of students seeing that the content and skills they are using in STEM experiences are applicable to the real world and will be useful in their current and future life in and beyond the classroom [25].

## Equity-Oriented STEM Literacy Framework

**Figure 2.** The Equity-Oriented STEM Literacy Framework [26] (p. 6).

Critical thinking and problem solving are at the heart of STEM literacy. This consists of the ability to use multiple disciplines to solve real-world problems and to engage in iterative problem solving as they overcome constraints to design the best solution to their problems [27]. They are listed in the standards of math practice [28], the science and engineering practices [29], and the technology and engineering practices [1], as well as the integrated STEM practices [27,30]

Dispositions towards STEM include interest in, motivation towards, and attitudes towards STEM. Student STEM dispositions impact choices in courses and careers and are therefore an important factor related to equity and access [31,32].

There is limited empirical research on the development of students of integrated STEM identities as teachers, and their identities as teachers of STEM in the K-12 setting. This might be due to the fact that the construct of identity has no one agreed upon definition [33]. To broadly define identity development, we used a combination of the ability to see the applicability and utility of subject [34] and the influence of parents, community, and peers [35,36].

Empathy is defined as the ability to understand what another person is experiencing [37]. Empathy is a significant component of the Equity-Oriented STEM Literacy Framework [25]. In the Equity-Oriented STEM Literacy Framework, empathy is the ability to identify with a person or a situation and not for someone else [25].

Empowering all learners to see themselves as change agents who are capable of learning and utilizing STEM principles and practices is an important part of providing meaningful and equitable instruction. This plays a role in their long-term persistence [30,38].

When integrated STEM education is centered around the Equity-Oriented STEM Literacy Framework, it can disrupt systems of oppression and privilege, while providing positive opportunities and access for all learners; this empowers them to be agents in systematic change. However, this can only happen if teachers understand the importance of these dimensions and create experiences that support the components.

### 1.4. Pre-Service Teachers and Dispositions

Positive experiences in integrated STEM education affect teachers' perceptions of STEM education [39]. Despite the many benefits of integrated STEM education, most U.S. schools continue to utilize single, isolated subject instruction [40]. The opposition to integrated STEM learning makes it difficult for PSTs to envision and enact effective STEM instruction. Due to the lack of meaningful STEM learning experiences, many PSTs perpetuate instruction that lacks meaningful, integrated, positive experiences for their students [41].

Often, the only experience that PSTs have with integrated STEM is in the teacher preparation courses [4]. However, integrated STEM pedagogies are only sometimes addressed in elementary methods classes [4]. Therefore, teachers are often hesitant to attempt facilitating integrated STEM learning experiences [42]. Teachers' current beliefs about integrated STEM and how to teach STEM subjects will be challenged as they learn to design and implement integrated STEM lessons [2].

The research indicates that thoughts of teaching STEM content evoke fear and anxiety among elementary pre-service and novice teachers [43] due to a lack of confidence and limited preparation in that area [44]. Learners may have negative STEM educational experiences due to the relationship between teachers' affective dispositions and the enacted teacher practices [45,46]. Educators' self-reported preparation [47] and confidence in STEM education [48] positively impact their STEM education practices. The attitudes and beliefs about teaching through integrated STEM are explored in this study. Dispositions are affected by perceptions, values, and ideas [49]. The enacted practices of teachers in the classroom are influenced by their dispositions [49–52].

PSTs have limited experience with MEAs, especially in elementary classrooms [19]. Because the PSTs' prior experiences influence their willingness to implement inquiry-based lessons, it is important to study how experiencing MEAs influences their dispositions towards and willingness to teach integrated STEM.

### 1.5. Theoretical Framework

The theoretical framework used for this study was situated learning theory. In situated learning theory, learning occurs best in a community of practice. This theory promotes the idea that true learning occurs when one experiences it authentically in contexts similar to those in which it will be applied [53]. Situated learning effectively explores affective dispositions and beliefs, as well as perceptions in specific learning situations [54]. Central to situated learning theory is the idea that interactions between the learner and the environment are mediated by social interactions [55,56]. By grounding the learning environment in the Equity-Oriented STEM Literacy Framework, we created a community of practice in which PSTs experienced and planned integrated STEM lessons using MEAs. Thus, in a community of practice, PSTs engaged in the high-quality learning activities of engaging in the authentic contexts of being a student, planning instruction, and teaching with a focus on equity and MEAs. The Equity-Oriented STEM Literacy Framework can be applied to PSTs engaging with integrated STEM in this setting (see Figure 3).

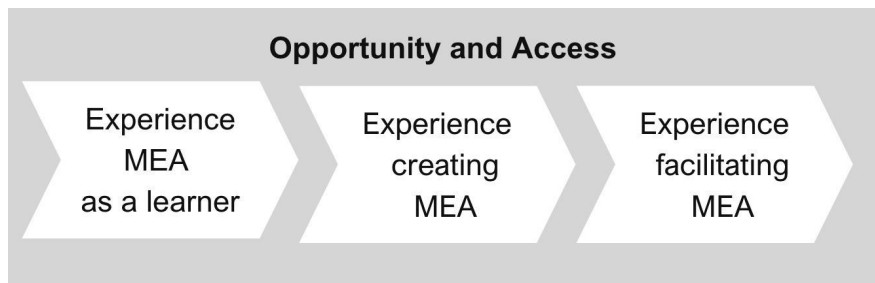

**Figure 3.** High-quality learning experience for PSTs.

For the PSTs, a high-quality integrated STEM learning experience involves engaging in MEAs as learners in their methods courses, then designing their own MEAs using the resources provided to them, and finally facilitating MEAs with the children who attended the informal STEM learning experiences.

Using STEM-situated learning theory, we examined how the PSTs' perceptions were influenced by participating in the high-quality STEM learning experience [54]. Similar applications of situated learning theory have investigated the relationships between PSTs' attitudes towards STEM in formal educational contexts [45].

## 2. Materials and Methods

### 2.1. Research Aims

This study aimed to deepen the knowledge base about PSTs' perceptions of an engineering-based approach to integrated STEM education through MEAs. It also examined how the experiences of PSTs relate to the K-12 Equity-Oriented STEM Literacy Framework. This study seeks to answer the following research question: How does experiencing MEAs influence PSTs' perceptions of an engineering-based approach to integrated STEM?

### 2.2. Participants

The research participants were 17 elementary (PSTs) from two public universities in the southeastern and western regions of the United States. Thirteen PSTs were White, three were Hispanic, and one was a Pacific Islander. Ten PSTs were taking their first methods course, and four were in the middle of their summer courses for future K-6 grade teachers who had participated in an informal STEM learning experience. For this project, pseudonyms are used to describe all the participants.

### 2.3. Settings

Informal STEM learning environments provide PSTs with a low-risk setting where they can authentically engage with children in the practices they are learning in their methods classes [57], and they have been used to introduce PSTs to integrated STEM [58]. This provided PSTs with a low-stakes environment where they could feel safe to try new pedagogical practices. Non-traditional placements like informal STEM learning experiences have been used to affect PSTs' dispositions in mathematics [59] and STEM education [60].

The methods classes were held via Zoom so that the instructors from the different universities could meet with all the PSTs. This allowed the PSTs to learn in a way that leveraged each instructor's expertise and perspective. The PSTs engaged in three MEAs designed around the theme of going to Mars. In one MEA, the PSTs were asked to design a spacesuit. The PSTs were asked to design a habitat structure that would withstand the cold temperatures, wind, and marsquakes in the second MEA. The final MEA required PSTs to build a communication tower using tape, paper, tinfoil, and scissors.

During this virtual meeting, the PSTs participated as students in one of the three MEAs. Afterwards, they were asked to create their own MEA with resources provided by the instructors. In groups, the PSTs facilitated their MEA with elementary-aged campers at the informal STEM learning experience.

### 2.4. Research Positionality

The three researchers who coded the reflection data were instructors of the elementary mathematics and science methods courses that the PSTs took before teaching in the informal STEM learning experience. These researchers engaged the PSTs in MEAs before asking them to write their integrated STEM lesson plans for the informal STEM learning experience. The MEAs designed through the lens of the Equity-Oriented K-12 Framework for STEM Literacy were implemented with the PSTs, so that they may experience them as learners. This allowed the PSTs to engage in tasks that were more open-ended than the traditional tasks they may have previously seen. The researchers firmly believe that experiences with

MEAs and integrated STEM learning are crucial to preparing PSTs to effectively engage learners in meaningful learning experiences.

### 2.5. Data Sources

The data consisted of class assignments, including reflections collected prior to and at the conclusion of course participation to discuss their experiences in the program. Sample items in the reflection included: "How confident do you feel teaching integrated STEM?", "Describe an impactful activity you experienced during camp? How did it impact you?", and "Do you feel more prepared to and/or more confident about teaching and creating lessons that integrate STEM disciplines after experiencing STEM activities in this course?" The reflections were used for coding and analysis; then, the data were informally triangulated by the teacher's observational notes and a debrief with the PSTs after they taught the MEAs in the informal learning experiences.

### 2.6. Analysis

A naturalistic inquiry explored how experiencing MEAs influenced the PSTs' perceptions of an engineering-based approach to integrated STEM. The Equity-Oriented K-12 Framework for STEM Literacy was used to develop the coding scheme [25]. We focused on the lived experience of the participants as they engaged in, wrote, and taught integrated STEM MEAs in an informal STEM learning environment [61]. Through their involvement, we were offered insights into the experiences of the PSTs and how these experiences shaped their understanding of an engineering-based approach to integrated STEM [62]. Explicit rules were followed while analyzing the reflections [63]. The researchers used deductive analysis and coded the reflections for evidence of the elements of the Equity-Oriented STEM Literacy Framework: *dispositions, applicability and utility, empowerment, critical thinking and problem solving, identity development,* and *empathy.*

### 2.7. Trustworthiness

Guba and Lincoln [64] established four elements of qualitative trustworthiness that researchers should abide by to ensure their research efforts are valid. These elements of trustworthiness consist of *credibility, dependability, confirmability,* and *transferability* [65], and this study achieved each element of trustworthiness. The researchers were committed to logically aligning the data collection methods to the goals of the research questions, which indicates credibility [66]. The dependability requirements were fulfilled through the use of inter-rater and intra-rater reliability [67,68]. For this study, both reliability ratings surpassed the minimum 90% agreement threshold [62]. This means that the level of agreement between the researchers was excellent and met the required standards for reliability analyses. To ensure confirmability, the researchers clearly outlined their positionality in this study, and they used memoing to monitor their reflexivity [67]. Transferability was achieved by providing an in-depth discussion of the research findings, as well as a rich description of the study's settings and participants in order to help readers envision how the findings could be applied to other situations [67,69,70].

## 3. Results

This section includes the results of our findings. We first considered how the Equity-Oriented K-12 Framework for STEM Literacy applied to the PSTs as STEM learners. We also considered how the PSTs observed the students engaging in the Equity-Oriented K-12 Framework for STEM Literacy. The results are organized according to the elements of the Equity-Oriented K-12 Framework for STEM Literacy, which include the applicability and utility of STEM, critical thinking and problem solving, dispositions, identity development, and empowerment (see Table 2).

**Table 2.** Results of coding using the Equity-Oriented K-12 Framework for STEM Literacy.

| Framework Category | Coding Frequency | % |
|---|---|---|
| Applicability and utility of STEM | 103 | 22 |
| Critical thinking and problem solving | 77 | 16 |
| Dispositions | 81 | 17 |
| Identity development | 178 | 37 |
| Empowerment | 39 | 8 |
| Empathy | 0 | 0 |

*3.1. Applicability and Utility of STEM*

Several PSTs recognized the utility and applicability of integrated STEM for teaching. Sarah reported using integrated STEM MEAs in her future classroom because "the camp opened [her] eyes to the ways in which STEM can be included within everything you do". Sarah also described integrated STEM as "a great tool for teachers to use to effectively teach skills in science, technology, engineering, and mathematics". Another PST reported feeling "prepared because I have observed that STEM is everywhere" and continued to describe "any connection made to real life is a connection to STEM".

Sydney reported believing that integrated STEM is "very exciting and interesting," but admittedly not something she thinks about daily. For Sydney, integrated STEM is "fun to sort of 'escape' into that realm, a very real realm". She also enjoyed seeing "how the products of these disciplines impact our daily lives" and enjoys "seeing [integrated STEMs] application every day" and wants to help "students make that same connection".

Maria said she "always valued the importance of students discovering what they love and encouraging them to explore future career endeavors. STEM lessons provide just that, so it is my job to make these values a reality through my teaching". Integrated STEM aligns with beliefs Maria held prior to the summer experience, and it became another tool she could use to support her students' natural curiosities. Maria found integrated STEM to be an essential tool for teaching mathematics and science content because students needed "a deep understanding of these concepts [and integrated STEM] encourages students to value their own thinking and find strategies that work for their learning". For Kaitlyn, implementing the MEAs showed her that math and science are "in everything and can be easily integrated in other lessons".

Participating in the summer learning experience influenced how the PSTs see the applicability and utility of STEM education. For some PSTs, STEM education was a natural extension of their beliefs and became an additional tool with which to engage students. Others realized that STEM was connected to everything and that if they were to teach any real-world context, they would default to including integrated STEM. Some were able to better see how mathematics and science could be implemented into lessons.

While engaging with elementary-aged students, the PSTs could see how the students could see the applicability and utility of STEM. Olivia stated that integrated STEM allowed the students to "envision themselves as future scientists or engineers" and let them take "ownership of their learning as they become the mathematician, engineer, or scientist that is tasked with solving a problem . . .. delv[ing]e into problems with a connection to the real world, making it relevant and engaging to students". Ellie found the habitat shelter MEA an engaging experience. Ellie stated, "[the students] had to relate to their own real-life experiences of a shelter and compare their lives on Earth to how life would be on Mars". For Ellie, the connection to the real world made the activity more relevant and engaging for the students.

Amelia described integrated STEM lessons as "ones that encourage student interaction and allow them to apply their learning to real-life situations". After engaging in the MEAs as learners and teachers, Amelia stated,

[Her] key takeaway from this course is that integration is key—meaning to fuse multiple subjects together, especially subjects that are more structured with subjects that

are more creative, and to present the content as applied to real-world circumstances that are relevant to the student's lives.

Initially, Amelia thought "STEM should be presented [in a] very structured and straightforward" manner. However, after the summer experience, she realized that STEM "(and should) be presented through storytelling and inquiry-based projects". She also realized the power of students seeing the applicability and utility of STEM, "STEM Camp also helped me realize the power of bringing in professionals who use Math for their jobs" because it "makes those jobs seem more attainable and proves the importance of math". In her future class, Amelia plans "on incorporating video calls from people in my network who work in areas such as aerospace, graphic design, finance, architecture, and more" since it "adds credibility to the lesson and motivates students".

While the PSTs initially recognized that STEM is connected to the real world, these experiences helped them see how STEM education connects real-world meaning to student learning and how it can be used to help students see the purpose of the standards they are learning in school.

### 3.2. Critical Thinking and Problem Solving

Some PSTs who participated in the summer experience found integrated STEM pedagogies different from those they had experienced in school. Ellie was able to see the value of creating activities where students could engage in critical thinking and problem solving. She said, "I see the value in STEM instruction. From my observations of the students at camp, it is evident that [the students] are highly engaged with the content". Her experience at STEM camp differed from her childhood experience because it made her think "science [does] not seem as intimidating as I had experienced as a child". She also saw STEM as a tool to encourage students to use critical thinking skills: "I also realized the importance of asking assessing and advancing questions which pushed students to think critically," and she saw "the value in integrated disciplines and how math and science can be educational and fun". She did not experience this kind of learning as a child because she was asked "only comprehension questions, but not questions that pushed me to think and evaluate my thought process". Sarah also saw the value in assessing and advancing questions because they "push[ed] [students] to dig deeper into concepts".

Seeing the students actively engage in critical thinking and problem solving shifted how other PSTs thought STEM was taught. Olivia noted that before working with the students, she thought "[the PSTs] would really have to guide the students with procedures and steps for solving a problem". After seeing the MEAs in action, she realized that as a teacher all she needed to do was "introduce the problem or task at hand" to students, then "provide them with directions and the materials" and provide the space for students "to design and create awesome things". Olivia also noted that the students were "able to problem solve and brainstorm things on their own, without so much teacher involvement," which allowed for "more active learning" to occur.

Experiencing MEAs as learners and teachers allowed the PSTs to see the power of empowering learners to solve problems. Several noted that the experience illuminated the need to let the students do the critical thinking because teachers often limit potential by not providing that space.

### 3.3. Dispositions, Identity Development and Empathy, Empowerment

According to the Equity-Oriented K-12 Framework for STEM Literacy, as PSTs and students engage in high-quality STEM activities their dispositions towards STEM, the development of their STEM identities, and their feelings of empowerment are all influenced simultaneously [25].

#### 3.3.1. Dispositions

For some PSTs, participating in the informal STEM learning experience changed their dispositions towards STEM. Before participating in the program, Chiara thought, "STEM

instruction meant almost every lesson would include like programming or a form of coding". She also reported being intimidated by STEM instruction because she thought "STEM subjects involved very complex concepts". However, after participating in the experience, she "realized how relevant STEM is to even simple daily tasks such as changing batteries in a device and using problem-solving to complete a task". Chiara also noted that the "STEM camp really showed me how creative and fun teaching skills like critical thinking as well as hard facts about Mars can be".

Before Maria taught her lesson, she also felt that she "was failing to assist [students]" because she could not answer their questions. After teaching, she felt more prepared to help "students who may feel stuck or confused about a STEM lesson". She further realized that as a teacher, "it [was] okay to not have all the answers" and that she just needed to "have questions available to prompt students who may feel stuck" due to the fact that " STEM lessons teach students the importance of critical thinking".

Before the experiences, Lucia reported thinking, "only individuals that are really good at math can teach science". After experiencing the MEA, Lucia's disposition changed, as she explained, "being part of STEM camp completely changed that idea. I know that I can easily teach science and make it super fun by adding different components". Lucia also noted that she "like[d] creating things, and that's a huge part of engineering!".

For these PSTs, participating in the summer learning experience positively influenced their dispositions towards STEM. Before the experience, the PSTs had a negative disposition towards STEM. Afterwards, they both viewed STEM in a more positive light.

### 3.3.2. Identity Development

All the PSTs' identities as STEM teachers were positively influenced by their participation in the summer program. Several initially reported that teaching STEM would be overwhelming, but after experiencing the summer program, they felt better prepared to teach integrated STEM. Brianna noted, "[she] thought of a STEM lesson as some unachievable lesson that required tons of resources and planning. I now know that a STEM lesson is very achievable and can be simple yet still valuable". Laura reported feeling "much better about teaching integrated STEM lessons to my students when I start teaching". Her experience during the summer "opened [her] eyes to the ways in which STEM can be included within everything you do" and that STEM did not need to be "a complex lesson". Another PST noted that they felt "confident because [they] now know what a STEM lesson involves and how to execute it well".

For Ellie, participating in the informal STEM learning experience provided "more confidence in diving in with creating STEM lesson plans and collaborating with other teachers in the process. She also said that the experience "framed [her] understanding of how [she] can integrate disciplines and create a STEM unit lesson plan that would engage students and target the Standards for Mathematical Practice".

Before the summer program, Maria also "felt it would be more overwhelming to create a STEM lesson and integrate it into the curriculum". Now, she sees the importance of STEM because it is "hands-on, exciting, and appealing to all students. It is important that people understand the world around them and how things work". She learned from participating in the summer experience "how important using the right language is, that encourages students to think critically, ask questions, and express their ideas freely". Maria expressed excitement about including integrated STEM in her future classroom. Maria stated, "There are so many tools and opportunities I can use to integrate a STEM lesson into the classroom that I did not see before, and now I am so excited to try them!". This indicated a shift from her prior belief that implementing integrated STEM would be overwhelming to a feeling of excitement.

As the PSTs' identities as STEM teachers were positively influenced by participating in the summer learning experience. Many of the PSTs reported a positive change in their dispositions towards integrated STEM as well. Before the experience, the PSTs reported being overwhelmed or unconfident in their abilities to create integrated STEM lessons.

Many PSTs' identities evolved through this experience. They saw themselves as being able to successfully implement STEM lessons because they embraced the idea of being facilitators of STEM lessons.

### 3.3.3. Empathy

Empathy is one of the elements of the Equity-Oriented K-12 Framework for STEM Literacy and was identified as one of the categories to be used in the deductive coding scheme. Although evidence of empathy was one of the codes identified by the researchers, it was not identified within any of the documents by the PSTs and therefore was not coded.

### 3.3.4. Empowerment

The PSTs reported feeling empowered by the summer learning experience. Audrey felt more confident because she could implement the teaching strategies she learned in her methods course. This experience was her first time teaching children, and she felt empowered as a teacher when the students loved participating. She said that "encouraged [her] to be a better teacher because I knew [the students] were hanging on to [her] every word", and that she would "never forget that initial welcome and will remember that feeling when [she has her] own classroom". This experience empowered Audrey as a teacher because of the experience she gained. It motivated her to use integrated STEM in the future.

The summer experience made Laura more confident in her ability to teach integrated STEM. She said, "teaching STEM is not as daunting," and she was confident she could "bring STEM into [her] classroom". Patricia felt empowered because she made a difference for a child when she helped the child "realize how art is important and incorporated with STEM" and increased the child's participation in the activity. The PSTs were empowered to be STEM teachers when the students developed positive STEM identities and dispositions towards STEM.

## 4. Discussion

In this section, we discuss the findings related to MEAs and the prior research conducted with PSTs. We end each section by describing how the findings connect to the Equity-Oriented STEM Literacy Framework.

### 4.1. Applicability and Utility of STEM

Throughout the final reflections completed by the PSTs, there were several references to real-world connections, which aligns with the Equity-Oriented STEM Literacy Framework [26]. Engaging students with problems that include real-world connections creates authentic learning opportunities, which is a crucial element of the PBL model [10,71]. The PSTs also made mention of self-discovery for students, whether that be their enjoyment of the STEM-related activities or the different approaches to problem solving that worked better for them than the typically prescribed methods. This type of self-authenticity is an important element of various equity frameworks, such as culturally relevant pedagogy [72]. In addition to PSTs reflecting on the authentic learning opportunities, there were also multiple indications of participants gaining insight into the impact that they could have on students both in and out of the classroom. These realizations suggest a connection to *the ethic of caring*, as defined by Ladson-Billings [72], in which teachers do not necessarily care for their students in an affectionate sense but instead by being fully cognizant of "the implication their works have on their students' lives, the welfare of the community, and unjust social arrangements" [72] (p. 474). Finally, the PSTs also indicated an awareness of students taking control of their own learning. This suggests an observation of self-directed learning tendencies that are necessary elements of successful PBL implementation [73]. Overall, the findings of this study suggest that the process of learning and teaching through MEAs conceptually aligns with the ideas of PBL as well as the multiple frameworks aimed at supporting equitable education.

The PSTs saw the applicability and utility of STEM both as learners and teachers of STEM. This finding aligns with the Equity-Oriented STEM Literacy Framework [25]. As the PSTs saw how STEM applied to the real world, their identities as STEM learners were developed; they could envision themselves as STEM teachers and were empowered to use integrated STEM in their future classrooms.

*4.2. Critical Thinking and Problem Solving*

While discussing the roles of critical thinking and problem-solving skills, the participating PSTs often referred to the differences between the integrated STEM model and how they were taught as children. Multiple PSTs suggested that they previously saw teaching the various STEM disciplines as highly procedural or reliant on fact memorization. However, they gained an appreciation for MEAs because they noticed students relying on the types of skills that the PSTs had not experienced using in similar settings as young learners. When the primary responsibility was to serve as a facilitator, the PSTs saw the value of engaging students in critical thinking skills by asking questions aimed at assessing potential gaps or helping students advance their learning rather than simply checking for comprehension. These findings align with the previous literature that suggests that implementing PBL can increase student engagement in critical thinking and other 21st-century skills (i.e., communication) [74,75]. The responses indicate that the PSTs now view science as much more accessible than the perceptions they held based on their experiences as children. This is partly because they feel that when teachers assume a facilitator role, they do not risk inhibiting students' learning potential. The importance that the PSTs place on the facilitator role as it pertains to the use and development of critical thinking and problem-solving skills demonstrates an awareness of the necessity of allowing students to learn in a way that suits them best in a self-directed manner; all of these aspects are key elements of the successful implementation of PBL [71,73].

The PSTs could identify when the students attending the camp engaged in critical thinking and problem solving. Seeing the students engage in critical thinking and problem solving caused the PSTs' dispositions towards STEM to shift positively and allowed them to develop their identities as STEM teachers. However, none of the PSTs identified that they were engaging in critical thinking and problem solving despite engaging in MEAs, creating MEAs, and facilitating MEAs. While engaging in this process, the PSTs used critical thinking skills to solve problems that demonstrated the applicability and utility of STEM, which positively impacted their dispositions towards STEM and their identities as STEM teachers. This is in line with the Equity-Oriented STEM Literacy Framework [26].

*4.3. Dispositions, Identity Development and Empathy, Empowerment*

4.3.1. Dispositions

There were no instances of PSTs referring to student dispositions. However, this could be explained by several possible reasons. First, these were early teaching experiences for the PSTs. Therefore, the research suggests that they struggle with professional noticing and focus more on their actions [76,77]. Secondly, the PSTs had very short interaction times with these learners; so, they needed to have a relationship that might occur in a long-term setting in order to fully understand the dispositions of the learners, and multiple experiences might be needed to note shifts in dispositions among young learners.

Although the PSTs did not see a dispositional shift in the children who participated in the summer camp, their dispositions did positively shift. After experiencing MEA, the PSTs indicated a positive shift in their own dispositions towards integrated STEM. This finding aligns with previous research concerning problem-based learning and is strongly correlated with the concepts of PBL [73,78]; it was that found exposure to problem-based learning positively influenced PST self-efficacy [79]. This also aligns with previous work completed by the researchers [60].

Our findings align with the Equity-Oriented STEM Literacy Framework because when the PSTs were considered STEM learners, their dispositions towards integrated STEM

shifted when they saw the applicability and utility of STEM and their students engaged in critical thinking and problem solving.

### 4.3.2. Identity Development

The participants reported that before the summer experience, they felt overwhelmed by and/or lacked confidence in creating integrated STEM lessons. However, the PSTs' responses indicate that participating in the program positively affected their attitudes towards integrated STEM education. The results of this study are similar to those of previous studies [9,32,45,57,60]. The activities they engaged in during this summer helped the PSTs to develop their identities as STEM teachers. Martin and Jamieson-Proctor [48] also found that PSTs with a positive identity shift as STEM teachers reported they were more likely to implement PBL in their future classrooms. In [45], the authors had similar findings when implementing MEAs with PSTs. Throughout their reflections, the participating PSTs highlighted that they gained confidence after realizing how student-centered integrated STEM lessons should be. The extent to which the lessons were focused on being student-centered allowed the PSTs to feel they were more of a facilitator rather than just an imparter of knowledge. This aligns with the ideas of PBL, given that the strategy relies on students taking ownership of their learning, which becomes mainly self-directed when using the pedagogical model [10,71]. These realizations that the PSTs highlighted in their reflections show that, as a result of participating in the summer experience, they gained confidence and now feel capable and prepared to implement STEM lessons successfully. As PSTs engaged in the high-quality learning experience of experiencing an MEA as a student, writing an MEA using the topic they were assigned, and facilitating an MEA with students, their identities as STEM teachers were developed. These findings are consistent with the Equity-Oriented STEM Literacy Framework [25]. Research indicates that when students engage in high-quality integrated STEM activities, they see themselves as makers and doers of STEM [59]. This study suggests that when activities are designed to influence students' STEM identity development, the PSTs' identities as STEM teachers and learners also positively shift. As students, in this case, the PSTs were engaged in high-quality learning experiences and began to see themselves as belonging to the STEM community, in this case as STEM teachers. Despite the consistent positive outcome towards the PSTs developing their identities as STEM teachers, there is little to no research on whether these outcomes are sustained and translate into observable teaching practices by the PSTs when they are the teachers of record in their own future classrooms.

### 4.3.3. Empathy

Empathy was not observed in the data from the PSTs or for the PSTs identifying empathy among their students. For empathy to be coded for the PSTs, the PSTs needed to be able to identify with the problem and to be able to relate to it. Additionally, the PSTs noticing empathy in the students was defined as being when the PSTs noticed empathy among the students, which was different from the students being empathetic towards each other. The PSTs did not observe empathy in the students to whom they taught the MEAs. This lack of noting empathy by PSTs aligns with the prior research [32,79]

### 4.3.4. Empowerment

The responses from the PSTs provided a variety of reflections that align with the literature concerning PBL and equitable education practices. Two participating PSTs indicated that they felt encouraged or more confident in their ability to teach science through integrated STEM MEAs. This positive influence on the PSTs' beliefs about their own teaching abilities, especially those related to the STEM disciplines, aligns with the previous research in which PSTs were introduced to PBL [56,80].

Student engagement was another highlight of the PST reflections that were related to how the PSTs and learners were empowered by using MEAs. Increased student engagement is a hypothesized and observed impact of implementing PBL [56,73,80]. Therefore,

this study shows that using MEAs aligns with the conceptual and empirical literature surrounding PBL. Finally, Patricia highlighted the impact on a student's life caused by helping the student realize the connection between art and STEM, which draws on the previously mentioned ethic of caring [72]. The ethic of caring is a common theme in research related to equity-based theories, particularly culturally relevant pedagogy. Therefore, it is not a surprising observation for this study [79,81,82]. In Patricia's case, being a difference maker in a student's life served to empower her as a STEM educator.

According to the Equity-Oriented STEM Literacy Framework, individuals feel empowered to become change agents when they engage in high-quality STEM learning experiences [25]. In this instance, the PSTs engaged in the high-quality learning experience of engaging in an MEA, creating their own MEA, and facilitating an MEA for students. After this activity, they reported feeling empowered to teach integrated STEM in their future classrooms, which is a change from the traditional teaching philosophy for siloed mathematics and science. This change in teaching and perceptions is one small step towards fostering the societal change agents in learners that the Equity-Oriented STEM Literacy Framework seeks to foster. In addition, with PSTs recognizing the importance of the elements in the Equity-Oriented STEM Literacy Framework, the hope is that they will enact these in their classrooms to foster an environment with an attention to equity and access for all.

In [57], the author et al. found that different kinds of integrated STEM activities impacted students differently and that children's most transformative learning experiences were when they were asked to solve problems with empathy. Empathy is a necessary element in STEM education to help motivate students to see the power of STEM [83]. In [79], the author et al. found that students were motivated to engage in STEM to help others. In [60], the author et al. also found that empathy motivated middle-level students to pursue a STEM career. For this study, empathy was defined as teachers reporting elementary-age students expressing the desire to help another using integrated STEM. Despite the power of empathy found in previous studies with students, the PSTs did not report any students engaging in empathy. The authors of the Equity-Oriented STEM Literacy Framework found that integrated STEM activities that developed empathy helped students see themselves as belonging in STEM despite experiencing other barriers to participating in STEM [26]. For other students, empathy can serve as a conduit, helping them see the applicability and utility of STEM in finding a solution to the problem they are exploring.

This study found a connection between the Equity-Oriented STEM Literacy Framework and the activities that the PSTs experienced during the summer. The applicability and utility of STEM was the most obvious theme that the PSTs observed. They could see how they could use MEAs in their future classrooms and how students use real-world contexts to solve MEAs. While the PSTs did not identify themselves as engaging in critical thinking and problem-solving skills, they could see that the students were engaging in these skills while they solved the MEAs the PSTs facilitated at the camp. The researchers argue that the PSTs engaged in critical thinking and problem solving when they designed the MEAs using the resources that their instructors provided. Despite the importance of empathy in the Equity-Oriented STEM Literacy Framework, it was not observed in this study. In the future, the power of empathy needs to be explicitly taught to PSTs so that they can harness its power in their classrooms. The high-quality task that the PSTs engaged in for this experience helped to develop the PSTs' identity as STEM teachers and positively influenced their dispositions towards STEM while empowering the PSTs to use STEM in their future classrooms. A future study should include following the PSTs into the classroom to see if the findings from this study transfer to long-term changes in teaching practices.

## 5. Conclusions

This study illuminates the importance of providing elementary PSTs with positive STEM learning and planning experiences that are grounded in equitable practices. Moreover, using engineering as the way to integrate mathematics and science through MEAs

gave the PSTs experience with engineering and positively impacted their dispositions towards using engineering to teach integrated STEM to their future students. Providing PSTs with opportunities to experience, create, and facilitate these STEM learning experiences can be a way for teacher educators to support the shifts in perception that are needed.

The data indicate that by engaging in engineering-based integrated STEM learning experiences, PSTs increase their knowledge about the applicability and utility of STEM while simultaneously developing their identities as STEM teachers, which positively influences their dispositions towards STEM and empowers them to be teachers of STEM. It also describes the need for teacher educators to be more purposeful about the experiences so that PSTs can create better activities to engage their future students. Finally, it highlights the importance of providing PSTs with the opportunity to design and create integrated STEM MEAs and the need for more longitudinal studies to highlight the importance of these experiences.

A possible limitation to this study is presented by the modality of instruction. The PSTs received an atypical introduction to integrated STEM education due to the virtual learning environment. Even though their learning environment was not standard for integrated STEM education, the PSTs still reported a positive shift in their dispositions. Therefore, it would be beneficial in future research to conduct a similar study in different educational settings to determine whether the approach transfers into more traditional modalities.

Another limitation is that this study explored the experiences of PSTs during one summer semester based on their reflections. It did not employ other perspectives or data sources to confirm or refute what was shared in the reflections. A longitudinal study with multiple data points and sources could show whether the changes observed in this study remain consistent or change over time. This would also allow further study beyond perceptions that could analyze changes in teaching practices over time. In addition, this study involved an informal learning environment, which may have impacted the experiences of the PSTs and the learners they instructed. Further exploration of different contexts will help to build a more robust understanding of the use of MEAs to equitably teach and learn integrated STEM.

**Author Contributions:** Conceptualization, C.M., J.M. and M.B., T.R. and L.O.T.; methodology C.M., J.M., M.B., T.R. and L.O.T.; software, C.M., M.B. and T.R; validation, C.M., M.B., T.R. and L.O.T.; formal analysis, C.M., M.B. and T.R.; investigation, C.M., M.B., T.R. and L.O.T.; resources, C.M., M.B., T.R. and L.O.T.; data curation, C.M., M.B., T.R. and L.O.T.; writing—original draft preparation, C.M., J.M., M.B., T.R. and L.O.T.; writing—review and editing, C.M., J.M., M.B. and T.R.; visualization, M.B. and C.M.; supervision, C.M., M.B. and L.O.T.; project administration, C.M. and M.B. All authors have read and agreed to the published version of the manuscript.

**Funding:** This research received no external funding.

**Institutional Review Board Statement:** The study was approved by the Institutional Review Board of Auburn University (#20-496/May 2020), University of California: Long Beach (#20-406/May 2020).

**Informed Consent Statement:** Informed consent was obtained from all subjects involved in the study.

**Data Availability Statement:** The data presented in this study are available on request from the corresponding author. The data are not publicly available due to privacy issues, but all identifying information on the location and person will be removed when shared.

**Conflicts of Interest:** The authors declare no conflict of interest.

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
