# Peer review of "Model-Eliciting Activities: Pre-Service Teachers’ Perceptions of Integrated STEM"

_education, doi:10.3390/educsci13121247_

Round 1

Reviewer 1 Report

Comments and Suggestions for Authors

See attached Word document.

Comments on the Quality of English Language

The use of English language is high-quality. Minor grammatical and usage errors.

Author Response

Thank you for your feedback. Please see our comments in the attached document. 

Reviewer 2 Report

Comments and Suggestions for Authors

Author Response

Thank you for your feedback. Please see the responses to your feedback in the attached file.

Reviewer 3 Report

Comments and Suggestions for Authors

Dear author, 

Many thanks for the development of a coherent and relevant paper on STEM education and pre-service teachers initiatives. In the following lines you will find my recommendations to improve your paper: 

1) The first paragraph of the introduction repeats many sections of the abstract. Both sections need to be different (introduction does not need to mention the main results of the study). 

2) In 1.2, 'Problem-based-learning' is included as a core concept of this manuscript. However, authors do not deepen nexus between this term and MEA. I suggest to elaborate a deeply explained justification in this matter. 

3) Section 1.5 contains important information on how the access to STEM education enhances more inclusion and educational opportunities. I suggest to change its title, because "Theoretical framework" does not provide a coherent description of the elements that are discussed in this part of the paper. 

4) In the methodological section no explanation was provided on how the instruments were validated or the categories that were retrieved in order to deepen into the research problem. It is also important to mention how the analysis and interpretation procedures were held by multiple researchers. 

5) Results and their discussion deeply elaborate on how STEM education is related with PSTS and the potentialities on these topics to innovate on teaching-learning processes.

6) Conclusions are limited. They need to elaborate on how the research question was answered, together with the methodological limitations and proposals. Finally, it is important to mention the future topics to study and roadmaps for researchers and practitioners in this topic. 

Author Response

Please see the attachment for responses to the reviewer reports.

Round 2

Reviewer 1 Report

Comments and Suggestions for Authors

Thank you for your careful consideration of my comments and the substantial revisions to your original manuscript.  The characterization of an MEA as a curricular approach to project-based learning and the elaboration on the components of the Equity-Oriented STEM Literacy Framework as an aprior coding scheme are much improved.  The implications for developing societal change agents is especially strong. Your use of the STEM literacy framework as an analytic lens within situated learning theory is much clearer.

I offer responses to your decision about two of my concerns along with line-by-line comments.  

1.  I continue to encourage you to consider reporting descriptive statistics (e.g. code frequency) for your 17 participants to offer more insight than "few PSTs" or "most PSTs".  I agree that you are not aiming for generalizability in this manscript, but these minor changes seem appropriate since you have used apriori coding.

2.  My comments about the the use of the online learning spaces as a limitation of this study was not in reference to the quality of instruction in this modality. You can highlightthe benefits of this modality to communicate the transferrability of your approach to other ways of preparing preservice teachers. You might consider doing so in conclusions instead of limitations.

Line by line comments

Page 2, line 43 - Preview MEAs as a transition to Sections  1.1 and 1.2

Page 3, lines 125 - 136 - Formatting the six design principles in a table would increase readability.

Page 6, line 207 - You are using the framework for more than analysis in this study (reference sentence on designing MEAs on page 9). The descriptors of the framework components could be formatted as bullets to increase readability.

Page 7, line 239 - The meaning of "airport" is unclear.

Page 19, lines 249 - 250 Beginning this sentence with "In general" feels awkward - your results are at the framework component level. 

Comments on the Quality of English Language

With and number of track changes and major edits, it was challenging in places to evaluate style and grammar. I found incomplete sentences and missing punctuation but did not complete a careful review for all potential errors. These errors do not relate to the quality of the manuscript content.

Author Response

Thank you for your feedback. Please see our responses to the suggested revisions below. 

1. I continue to encourage you to consider reporting descriptive statistics (e.g. code frequency) for your 17 participants to offer more insight than "few PSTs" or "most PSTs". I agree that you are not aiming for generalizability in this manuscript, but these minor changes seem appropriate since you have used apriori coding.

  • We have added a table to page 8 with the code frequency and percentages.

2. My comments about the the use of the online learning spaces as a limitation of this study was not in reference to the quality of instruction in this modality. You can highlightthe benefits of this modality to communicate the transferrability of your approach to other ways of preparing preservice teachers. You might consider doing so in conclusions instead of limitations.

  • We revised the conclusion to include your suggestions. Please see lines 679-686

Line by line comments
Page 2, line 43 - Preview MEAs as a transition to Sections 1.1 and 1.2

  • We included a transistion sentence to better transision between sections 1.1 and 1.2

Page 3, lines 125 - 136 - Formatting the six design principles in a table would increase readability.

  • We have included the table as suggested.

Page 6, line 207 - You are using the framework for more than analysis in this study (reference sentence on designing MEAs on page 9). The descriptors of the framework components could be formatted as bullets to increase readability.

  • We have included a table with the framework, code count and percentage coded.

Page 7, line 239 - The meaning of "airport" is unclear.

  • Thank you for catching the typo. We have corrected the error. It now says support.

Page 19, lines 249 - 250 Beginning this sentence with "In general" feels awkward - your results are at the framework component level.

We removed the phrase in general, as suggested.

Reviewer 2 Report

Comments and Suggestions for Authors

Author Response

Thank you for your feedback. As requested, we have added a  table to include the codes and percentages. This table is on page 8.

On page 12, line 475, we have rewritten the statement to include the correct pseudonym. 

Reviewer 3 Report

Comments and Suggestions for Authors

This paper has addressed the observations made in the first round of revisions. Furthermore, it elaborates a coherent and congruent perspective on the role of STEM education in the formation of pre-service teachers.

Author Response

Thank you for reviewing our paper. We appreciate the time and feedback provided.